# SKILLBERT: "SKILLING" THE BERT TO CLASSIFY SKILLS!

## ABSTRACT

In the age of digital recruitment, job posts can attract a large number of applications, and screening them manually can become a very tedious task. These recruitment records are stored in the form of tables in our recruitment database (Electronic Recruitment Records, referred to as ERRs). We have released a de-identified ERR dataset to the public domain[1]. We also propose a BERT-based model, SkillBERT, the embeddings of which are used as features for classifying skills present in the ERRs into groups referred to as "competency groups". A competency group is a group of similar skills and it is used as a matching criteria (instead of matching on skills) for finding the overlap of skills between the candidates and the jobs. This proxy match takes advantage of the BERT's capability of deriving meaning from the structure of competency groups present in the skill dataset. In our experiments, the SkillBERT, which is trained from scratch on the skills present in job requisitions, is shown to be better performing than the pre-trained BERT (Devlin et al., 2019) and the Word2Vec (Mikolov et al., 2013). We have also explored K-means clustering (Lloyd, 1982) and spectral clustering (Chung, 1997) on SkillBERT embeddings to generate cluster-based features. Both algorithms provide similar performance benefits. Last, we have experimented with different machine learning algorithms like Random Forest (Breiman, 2001), XGBoost (Chen & Guestrin, 2016), and a deep learning algorithm Bi-LSTM (Schuster & Paliwal, 1997; Hochreiter & Schmidhuber, 1997). We did not observe a significant performance difference among the algorithms, although XGBoost and Bi-LSTM perform slightly better than Random Forest. The features created using SkillBERT are most predictive in the classification task, which demonstrates that the SkillBERT is able to capture information about the skills' ontology from the data. We have made the source code and the trained models[1] of our experiments publicly available.

## 1    INTRODUCTION

Competency group can be thought of as a group of similar skills required for success in a job. For example, skills such as *Apache Hadoop*, *Apache Pig* represent competency in Big Data analysis while *HTML*, *Javascript* are part of Front-end competency. Classification of skills into the right competency groups can help in gauging candidate's job interest and automation of the recruitment process. Recently, several contextual word embedding models have been explored on various domain-specific datasets but no work has been done on exploring those models on job-skill specific datasets.

Fields like medical and law have already explored these models in their respective domains. Lee et al. (2019) in their BioBERT model trained the BERT model on a large biomedical corpus. They found that without changing the architecture too much across tasks, BioBERT beats BERT and previous state-of-the-art models in several biomedical text mining tasks by a large difference. Alsentzer et al. (2019) trained publicly released BERT-Base and BioBERT-finetuned models on clinical notes and discharge summaries. They have shown that embeddings formed are superior to a general domain or BioBERT specific embeddings for two well established clinical NER tasks and one medical natural language inference task (i2b2 2010 (Uzuner et al., 2011), i2b2 2012 (Sun et al., 2013a;b)), and MedNLI (Romanov & Shivade, 2018)).

---

[1] https://www.dropbox.com/s/wcg8kbq5btl4gm0/code_data_pickle_files.zip?dl=0!

Beltagy et al. (2019) in their model SciBERT leveraged unsupervised pretraining of a BERT based model on a large multi-domain corpus of scientific publications. SciBERT significantly outperformed BERT-Base and achieves better results on tasks like sequence tagging, sentence classification, and dependency parsing, even compared to some reported BioBERT results on biomedical tasks.

Similarly, Elwany et al. (2019) in their work has shown the improvement in results on fine-tuning the BERT model on legal domain-specific corpora. They concluded that fine-tuning BERT gives the best performance and reduces the need for a more sophisticated architecture and/or features. In this paper, we are proposing a multi-label competency group classifier, which primarily leverages: SkillBERT, which uses BERT architecture and is trained on the job-skill data from scratch to generate embeddings for skills. These embeddings are used to create several similarity-based features to capture the association between skills and group. We have also engineered features through clustering algorithms like spectral clustering on embeddings to attach cluster labels to skills. All these features along with SkillBERT embeddings are used in the final classifier to achieve the best possible classification accuracy.

## 2 METHODOLOGY

As no prior benchmark related to job-skill classification is available, we manually assigned each skill in our dataset to one or more competency groups with the help of the respective domain experts to create training data. We experimented with three different models: pre-trained BERT, Word2vec, and SkillBERT to generate word embeddings. Word2vec and SkillBERT were trained from scratch on our skill dataset. We created some similarity-based and cluster-based features on top of these embeddings. Except for these features, some frequency-based and group-based features were also generated. A detailed explanation of all the steps is mentioned in the next sections. The details of dataset design and feature engineering used for model creation are given in the next sections.

### 2.1 TRAINING DATA CREATION

Our approach uses a multi-label classification model to predict competency groups for a skill. However, as no prior competency group tagging was available for existing skills, we had to manually assign labels for training data creation. For this task, the skill dataset is taken from our organization's database which contains 700,000 job requisitions and 2,997 unique skills. The competency groups were created in consultation with domain experts across all major sectors. Currently, there exists 40 competency groups in our data representing all major industries. Also within a competency group, we have classified a skill as *core* or *fringe*. For example, in *marketing* competency group, *digital marketing* is a *core* skill while *creativity* is a *fringe* skill. Once training data is created, our job is to classify a new skill into these 40 competency groups. Some skills can belong to more than one category also. For such cases, a skill will have representation in multiple groups. Figure 1 shows an overview of the datasets used in this step.

### 2.2 FEATURE ENGINEERING

For feature creation, we have experimented with Word2vec and BERT to generate skill embeddings. By leveraging these skill embeddings we created similarity-based features as well. We also used clustering on generated embeddings to create cluster-based features. As multiple clustering algorithms are available in the literature, we evaluated the most popular clustering algorithms – K-means (Lloyd, 1982) and spectral clustering for experimentation. We have done extensive feature engineering to capture information at skill level, group level, and skill-group combination level. The details of features designed for experiments are given below.

### 2.2.1 EMBEDDING FEATURES

Traditionally, n-gram based algorithms were used to extract information from text. However, these methods completely ignore the context surrounding a word. Hence, we have experimented with Word2vec and BERT based architecture to learn embeddings of skills present in training data. The details of how we have leveraged them in our problem domain are given below.

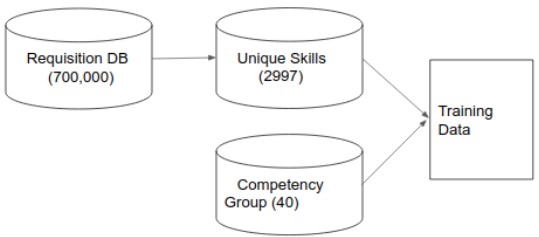

Figure 1: Dataset used for training data creation

**Word2vec** is one of the most popular techniques to learn word embeddings using a neural network. It uses a shallow, two-layer neural network to generate n-dimensional embedding for words. To use the Word2vec model on requisition data, we extracted skills from job requisitions and constructed a single document. Each sentence of this document represents the skills present in one requisition. As a requisition can have multiple skills, we created a 2-dimensional list, where the outer dimension specifies the sentence and inner dimension corresponds to the skills in that sentence. E.g. if there are two job requisitions called req1 and req2 and their corresponding skills are "Java, J2EE" and "Logistic regression, Data visualization, NLP" then outer index 0 corresponds to req1 and outer index 1 corresponds to req2. Index 0,0 will refer to Java and Index 0,1 will refer to J2EE and so on. Also before feeding this data for training lowercasing of words, stop word removal and stemming was performed as part of preprocessing. A total of more than 700,000 requisitions were used for model training. We have used embeddings of size 30 which was decided after evaluating model performance on different embedding sizes.

**BERT** Bidirectional Encoder Representations from Transformers, is designed to pre-train deep bidirectional representations from the unlabeled text by jointly conditioning on both left and right context in all layers. Pre-trained BERT model can be fine-tuned with just one additional output layer to create state-of-the-art models for tasks such as question answering, next sentence prediction, etc. Similar to Word2vec, BERT can also be used to extract fixed-length embeddings of words and sentences, which can further be used as features for downstream tasks like classification. But unlike fixed embedding produced by Word2vec, BERT will generate different embedding for an input word based on its left and right context. BERT has shown performance improvement for many natural language processing tasks. However, it has been minimally explored on the job-skill database. Hence, we leveraged BERT architecture on skill data to train the SkillBERT model. We have leveraged *ml.p2.xlarge* type *12 GIB* GPU memory *1xK80* GPU available on the AWS cloud for SkillBERT training and it took us around 72 hours to completely train it on our dataset. In the next section, we have given the details of training BERT on skill corpus.

**Training:** For training BERT, we used the same corpus as used for Word2vec training and experimented with hyperparameters like learning rate and maximum sequence length. For the learning rate, we used 0.1, 0.05, and 0.01 and finalised 0.01. For maximum sequence length, we used 64, 128, 180 and finalised 128. We could not perform extensive hyperparameter tuning due to hardware limitations. Once the training is finished, we extract the last hidden layer output of size 768 and further reduce the embedding size to 128 to decrease the training time of our final model discussed in the experiment section. For the dimensionality reduction of embedding, we did experiments with embeddings of size 32, 64, 128 and 256. As shown in Appendix Table 6, the best results were obtained using embedding of size 128. To make sure information from all the 768 dimensions is leveraged, we trained a 2-layer neural network classifier using SkillBERT embeddings as an independent feature and competency group as a dependent variable. Out of the 2,997 skills, 80% were used for training and the rest of the 20% were used for the validation. This model generates the probability values of a skill belonging to each of the 40 competency groups and was used as a feature in the final model at skill and competency group combination level. We have referred this feature as "bert-prob" in the rest of the sections. Figure 2 represents the architecture of the model used for getting these probabilities.

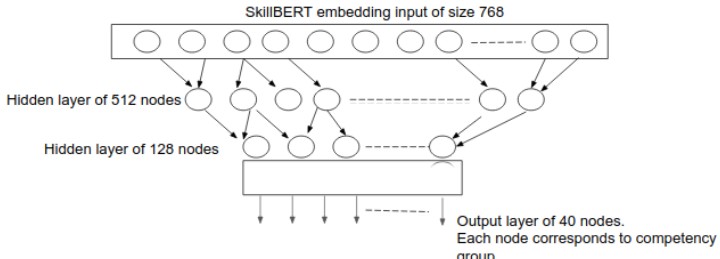

Figure 2: Classifier architecture

### 2.2.2 SIMILARITY-BASED FEATURES

By leveraging skill embeddings generated using embedding techniques, similarity-based features were created to capture the association between a group and skill. The details of those are given below.

**Similarity from competency group:** In competency group data, the name of each competency group is also present as a skill. We created a similarity score feature measuring the cosine distance between competency group name and skill embeddings.

**Similarity from top skills per group:** Apart from utilizing the similarity between competency group name and skill, we have also created similarity-based features between a given skill and skills present in the competency group. As an example, we have a skill named *auditing* and competency group *finance*. Three similarity-based features were created called top1, top2, and top3, where top1 is cosine similarity score between skill *auditing* and most similar skill from *finance*, top2 is the average cosine similarity score of top two most similar skills and top3 is the average cosine similarity score of top three most similar skills. As shown in Appendix Table 5, the use of similarity-based features beyond three skills did not improve model performance.

### 2.2.3 CLUSTER-BASED FEATURE

For generating labels for skills using clustering, we experimented with two techniques on SkillBERT embedding – *K-means* and *spectral clustering*. Scikit-learn package of K-means was used to generate 45 cluster labels. The number 45 was decided by using the elbow method, the graph of which is shown in Appendix Figure 8. Using spectral clustering we generated 35 cluster labels. The details of how we used spectral clustering on SkillBERT embedding to generate cluster-based feature are given in Appendix section A.1.

### 2.2.4 SKILL TFIDF FEATURE

TFIDF (Salton & McGill, 1986; Ramos, 1999) is widely used in text mining to find rare and important words in a document, and as in our training data a single skill can be part of multiple competency groups, we used the same strategy to find skills that are unique to a competency group by calculating their TFIDF value. However, as group information will not be available for new skills, we will calculate the TFIDF of such skills differently. First, we will find the most similar top 3 existing skills and thereafter, take the average of their TFIDF values. This resultant value will be the TFIDF value for a new skill.

### 2.2.5 CORE AND FRINGE SKILLS

Apart from the features mentioned in the above sections, we have also created group-based features by counting the number of core and fringe skills in each group.

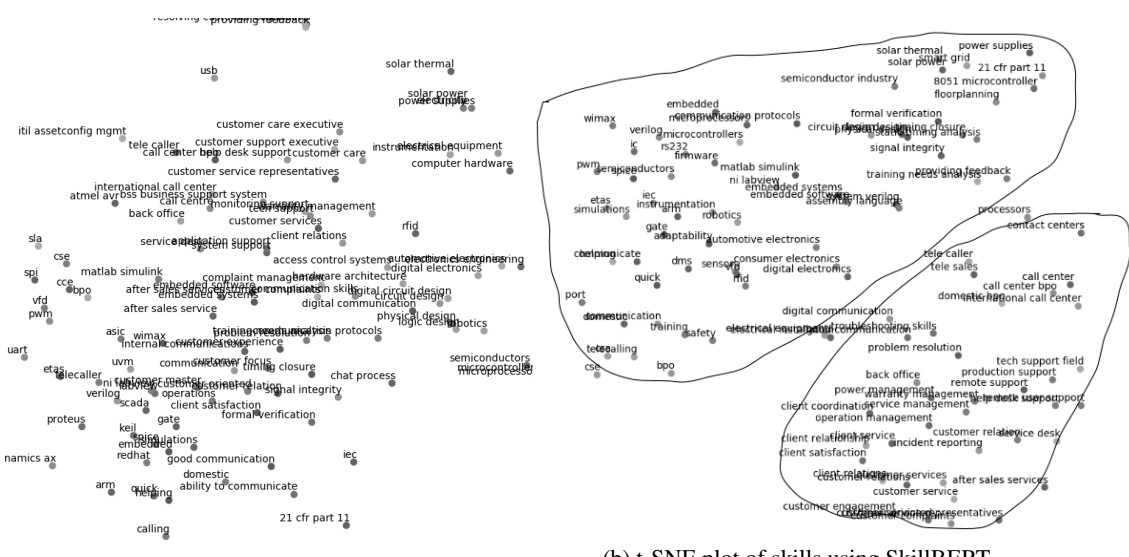

(a) t-SNE plot of skills using pre-trained BERT

(b) t-SNE plot of skills using SkillBERT

Figure 3: t-SNE plot of embeddings of "Customer Support" and "Electronics" competency group. The left image shows the projection generated using pre-trained BERT embedding and the right image is the SkillBERT plot. The top cluster shown in SkillBERT t-SNE plot represents "Electronics" competency group while the bottom cluster represents "Customer Support".

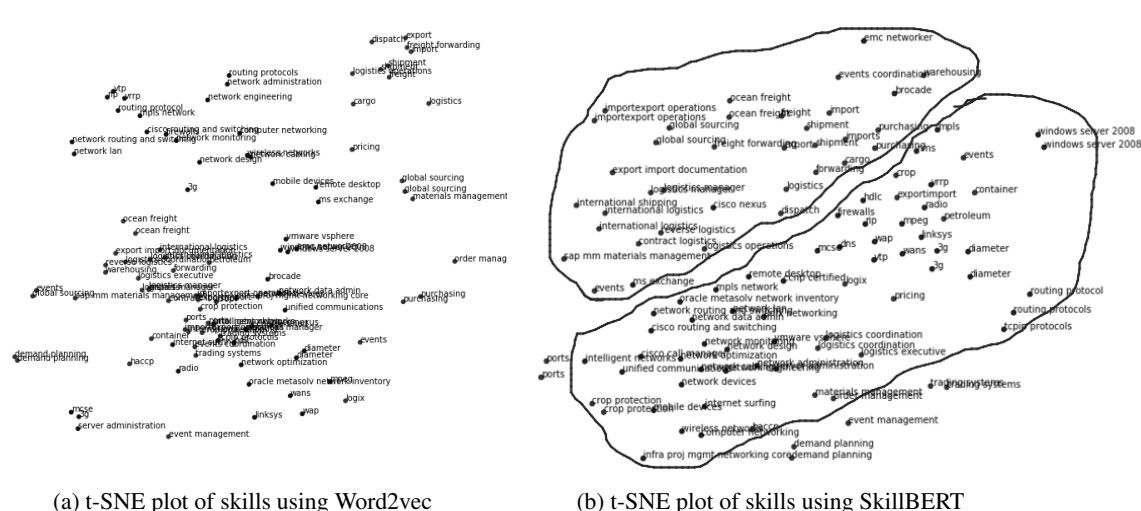

(a) t-SNE plot of skills using Word2vec

(b) t-SNE plot of skills using SkillBERT

Figure 4: t-SNE plot of embeddings of "Logistic" and "Network" competency group. The left image shows the projection generated using Word2vec embedding and the right image is the SkillBERT plot. The top cluster shown in SkillBERT t-SNE plot represents "Logistic" competency group while the bottom cluster represents "Network".

Table 1: Machine Learning model Hyperparameters

| Machine Learning Models | Best Hyperparameters | Hyperperameter bound |
|---|---|---|
| XGBoost | N_estimators:800, Depth:5 | N_estimators:400-1000, Depth:3-7 |
| Random forest | N_estimators:700, Depth:4 | N_estimators:400-1000, Depth:3-7 |
| Bi-LSTM | Layers:2(Nodes: 128, 64), Optimizer:Adam, Dropout:0.2 | Layers:2 - 4 , Nodes: 32 - 512, Dropout: 0.1 - 0.5 |

## 3 EXPERIMENTS

We have approached the categorization of skills into multiple competency groups as a multi-label classification problem. We have created our training data at *skill X competency group* level i.e. for each skill we will have 40 rows, corresponding to each competency group. For each skill-group pair, we have tried to predict the probability of that skill belonging to that group using classifier models like XGBoost, Random Forest, and Bi-LSTM. Pairs of models which were compared and had a statistically significant difference in the performance are highlighted with a star in Table 3. The details of all these experiments are given below.

**SkillBERT vs Word2vec vs Pre-trained BERT:** As the first experiment, we did a comparative study among SkillBERT, pre-trained BERT, and Word2vec models. For pre-trained BERT, we used the "bert-base-uncased" model which also produces embeddings of size 768. Similar to SkillBERT, we reduced embedding size to 128 and generated "bert-prob" feature. All features except cluster labels discussed in the feature engineering section were created using these embedding models. To better analyze the quality of embeddings, we projected high dimensional embeddings of skills present in competency groups in 2-D using t-SNE (van der Maaten & Hinton, 2008). From visualization shown in Figure 3 and Figure 4, it is clear that SkillBERT embeddings reduced the overlapping gap between groups and gave well-defined cluster boundaries as compared to word2vec and pre-trained BERT. As a classifier, we used XGBoost and performed hyperparameter tuning through grid-search to get the best possible result without over-fitting. In the training dataset, there was a total of 95,904 records and 2,398 unique skills while the testing dataset had 23,976 records and 599 unique skills. The results of this experiment are given in Table 3.

Table 2: Machine Learning model training time

| Machine Learning Models | Training Time (in seconds) |
|---|---|
| XGBoost | 122 |
| Random forest | 87 |
| Bi-LSTM | 167 |

**K-means vs spectral clustering:** In this experiment, we tried to see the effect of adding cluster-based features generated using K-means and spectral clustering on SkillBERT embedding. For this comparison, we applied XGBoost on the cluster labels and the features used in the previous experiment where we compared different embedding approaches.

**Random Forest vs Bi-LSTM vs XGBoost:** As part of this experiment, we applied Bi-LSTM, Random Forest, XGBoost and spectral clustering based features on SkillBERT and compared their performance. Table 1 contains the best performing hyperparameter values and their variation range during tuning through grid-search for all the classifiers used. The number of hyperparameter search trials done was 20, 20, 36 for XGBOOST, Random Forest, and Bi-LSTM models respectively. Table 2 contains the training time of each classifier model.

**Core vs fringe skill classification:** Finally, we also trained a multi-label and multi-class classifier to see how accurately we can classify *core* and *fringe* skills. For this, we trained a model with 3 classes where, class 0 – *no label*, class 1 – *fringe skill*, and 2 – *core skill*. All the features used in the last experiment were leveraged for this experiment and Bi-LSTM was used as a classifier.

**Impact evaluation:** By matching competency groups instead of skills, we are broadening the spectrum for matching the skills between candidates and jobs. For instance, if a candidate has mentioned skills like 'AngularJS', 'CSS', and 'JavaScript', then there is a high probability that the

candidate knows 'HTML' too because all these skills belong to 'Web development' competency group. Matching using competency groups takes care of similar cases by providing an aggregation to the skills typically required for matching the skills between candidates and jobs. While screening the candidate resumes hiring managers come across many skills which are unknown to them. By normalizing the skills to the competency groups using SkillBERT, we are reducing the time taken by the hiring managers to find the domain of the skills. The difference in time is because the SkillBERT not only matches the skills to their domains (groups) but also shows constituent skills in each group, thereby providing more context about the groups. As of now, there is no automated way of tracking the resume screening rate on our platform. However, post introduction of SkillBERT, a 150% increase has been observed in the number of average resumes screened per day. The above metric does not account for the confounders like hiring manager's experience and performance among other covariates.

Table 3: Evaluation of result on different embedding models and feature set

| Model | Precision | | Recall | | F1-score | |
|---|---|---|---|---|---|---|
| | Class 0 | Class 1 | Class 0 | Class 1 | Class 0 | Class 1 |
| *XGBoost + pre-trained BERT | 98.83% | 51.54% | 95.85% | 74.26% | 97.21% | 60.84% |
| *XGBoost + Word2vec | 98.06% | 68.34% | 97.36% | 65.21% | 96.53% | 66.73% |
| XGBoost + SkillBERT | 99.32% | 96.65% | 99.47% | 84.82% | 99.39% | 90.35% |
| XGBoost + SkillBERT + K-means | 99.27% | 96.92% | 99.54% | 85.24% | 99.40% | 90.70% |
| Random forest + SkillBERT + spectral clustering | 99.28% | 95.15 % | 99.50% | 83.48% | 99.39% | 88.93% |
| XGBoost + SkillBERT + spectral clustering | 99.35% | 97.23% | 99.48% | 85.09% | **99.41%** | **90.76%** |
| *Bi-LSTM + SkillBERT + spectral clustering | 99.26% | 95.86% | 99.57% | 86.43% | **99.42%** | **90.90%** |

# 4 RESULTS

Results shown in Table 3 conclude that SkillBERT improved the performance of the classification model over Word2vec and pre-trained BERT. Use of XGBoost with SkillBERT based features give F1-score of 90.35% for class 1 as compared to 60.83% and 66.73% of pre-trained BERT and Word2vec based features. Use of different machine learning (XGBoost and Random Forest), deep learning (Bi-LSTM) algorithms, and clustering-based features (K-means and spectral clustering) on top of SkillBERT is not making a statistically significant difference and the results are very similar. The difference between the validation dataset and test dataset F1 scores was less than 0.65 and 0.5 percentage points and the variance of validation data F1 scores for different hyperparameter trials was 1.20 and 1.05 percentage points for XGBoost+SkillBERT+spectral clustering and Bi-LSTM+SkillBERT+spectral clustering models respectively. We computed feature importance using the XGBoost model and "bert-prob" explained in section 2.2.1 created using SkillBERT was the top feature in the list. TFIDF and similarity-based features were also highly predictive. Next, the results of experiment 4 (core vs fringe skill classification) given in Table 4 show that we were able to classify fringe skills for a group more accurately compared to core skill. All the reported results are statistically significant at $p < 0.05$.

Table 4: Core vs fringe skill classifier results

| Precision | | | Recall | | | F1-score | | |
|---|---|---|---|---|---|---|---|---|
| Class 0 | Class 1 | Class 2 | Class 0 | Class 1 | Class 2 | Class 0 | Class 1 | Class 2 |
| 99.07% | 93.19% | 99.76% | 99.74% | 78.28% | 62.45% | 99.40% | 85.08% | 76.81% |

## 5 CONCLUSION

In this paper, we have addressed the problem of recruiters manually going through thousands of applications to find a suitable applicant for the posted job. To reduce the manual intervention, a multi-label skill classification model is developed which can classify skills into competency groups and hence, helps in quick mapping of relevant applications to a job. For skill representation, different word embedding models like Word2vec and BERT are used and comparison among classification results of different machine learning models is shown. Additionally, features like TFIDF, clustering labels, and similarity-based features are explored for better classification of skills. We trained BERT on a domain-specific dataset and a significant improvement is noticed while comparing the results with pre-trained BERT and Word2vec.

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

## A APPENDIX

### A.1 SPECTRAL CLUSTERING

Spectral clustering is a widely used unsupervised learning method for clustering. In spectral clustering, the data points are treated as nodes of a graph and these nodes are then mapped to a low-dimensional space using eigenvectors of graph laplacian that can be easily segregated to form clusters. Spectral clustering utilizes three matrices, details of those are given below.

**1. Similarity graph (Affinity matrix):** A similarity graph is a pair G = (V, A), where V={$v_1$,....,$v_m$} is a set of nodes or vertices. Different skills are forming different nodes as shown in Figure 5. A is a symmetric matrix called the affinity matrix, such that $ba_{ij} \geq 0$ for all i,j $\in$ {1,.......,m}, and $ba_{ii} = 0$ for i = 1,.....,m. We say that a set{$v_i$,$v_j$} is an edge if $ba_{ij} > 0$. Where $ba_{ij}$ is bert affinity between nodes i and j computed using cosine similarity between SkillBERT embeddings of the corresponding skills. The corresponding (undirected) graph (V,E) with E = {{$v_i$,$v_j$} | $ba_{ij}$ >0}, is called the underlying graph of G. An example of similarity graph structure as affinity matrix is shown in Figure 5.

**2. Degree matrix(D):** If A is an m×m symmetric matrix with zero diagonal entries and with the other entries $ba_{ij} \in$ R arbitrary, for any node $v_i \in$V, the degree of $v_i$ is defined as

$$d = d(v_i) = \sum_{j=1}^{m} |ba_{ij}| \qquad (1)$$

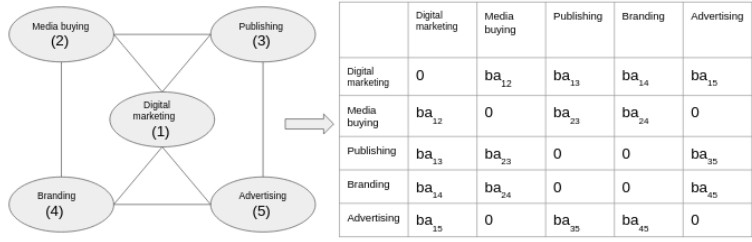

Figure 5: Adjacency matrix representation of Graph

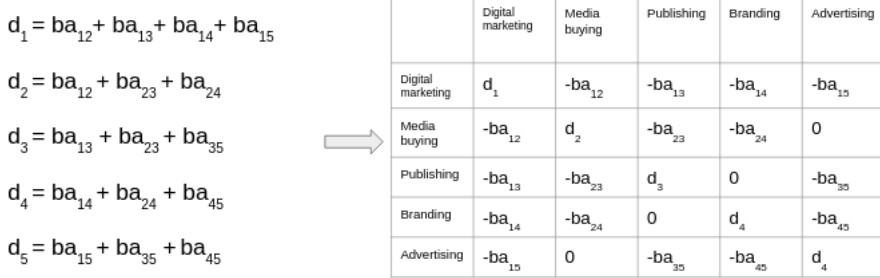

Figure 6: Graph laplacian for example in Figure 5

and degree matrix D as

$$D = diag(d(v_1), .........., d(v_m)) \tag{2}$$

**3. Graph laplacian (L):** If D is a diagonal matrix and A is affinity matrix then we can compute L as follows :-

$$L = D - A \tag{3}$$

The Laplacian's diagonal is the degree of our nodes, and the off-diagonal is the negative edge weights (similarity between nodes). For clustering the data in more than two clusters, we have to modify our laplacian to normalize it.

$$L_{norm} = D^{-1/2}LD^{-1/2} \tag{4}$$

We know that

$$L_{norm}X = \lambda X \tag{5}$$

Where X is the eigenvector of $L_{norm}$ corresponding to eigenvalue $\lambda$. Graph Laplacian is a semi-positive definite matrix and therefore, all its eigenvalues are greater than or equals to 0. Thus, we get eigenvalues $\{\lambda_1, \lambda_2, ..., \lambda_n\}$ where $0 = \lambda_1 \geq \lambda_2 \geq ... \geq \lambda_n$ and eigenvectors $X_1, X_2,...,X_n$. An example of a sample laplacian matrix is given in Figure 6. Once we calculate the eigenvalues of $L_{norm}$ and eigenvectors corresponding to smallest k eigenvalues where k is number of clusters, we create a matrix of these eigenvectors stacking them vertically so that every node is represented by the corresponding row of this matrix and use K-means clustering to cluster these new node representations into k clusters. For our experiment, we chose the first 35 eigenvectors to create 35 clusters and used them as features for model training. The number 35 was decided using the criteria of difference between two consecutive eigenvalues. As shown in Figure 7, the difference between eigenvalue 35 and 36 is significantly bigger.

## A.2 MISCELLANEOUS

This section contains the results of experiments done for hyperparameter selection and some figures referenced in the main text. Figure 8 Shows the elbow method graph for deciding the number of clusters in K-means. Table 5 shows the results of experiments done for a varied number of top skills for similarity based features. Table 6 shows the effect of different SkillBERT embedding size on the results of XGBoost classifier.

Table 5: Result for different Number of top skills similarity values in feature set (In this experiment, all the features mentioned in the experiment section "SkillBERT vs Word2vec vs Pre-trained BERT" were used and only the number of skills used for similarity value calculation were varied. As a classifier we used XGBoost)

| No. of skills used | Precision | | Recall | | F1-score | |
| --- | --- | --- | --- | --- | --- | --- |
| | Class 0 | Class 1 | Class 0 | Class 1 | Class 0 | Class 1 |
| Top 1 skill | 99.22% | 95.15% | 98.89% | 83.92% | 99.05% | 89.18% |
| Top 2 skills | 99.27% | 96.10% | 99.26% | 84.10% | 99.26% | 89.70% |
| Top 3 skills | 99.32% | 96.65% | 99.47% | 84.82% | 99.39% | 90.35% |
| Top 4 skills | 99.21% | 95.56% | 99.40% | 84.69% | 99.30% | 89.80% |

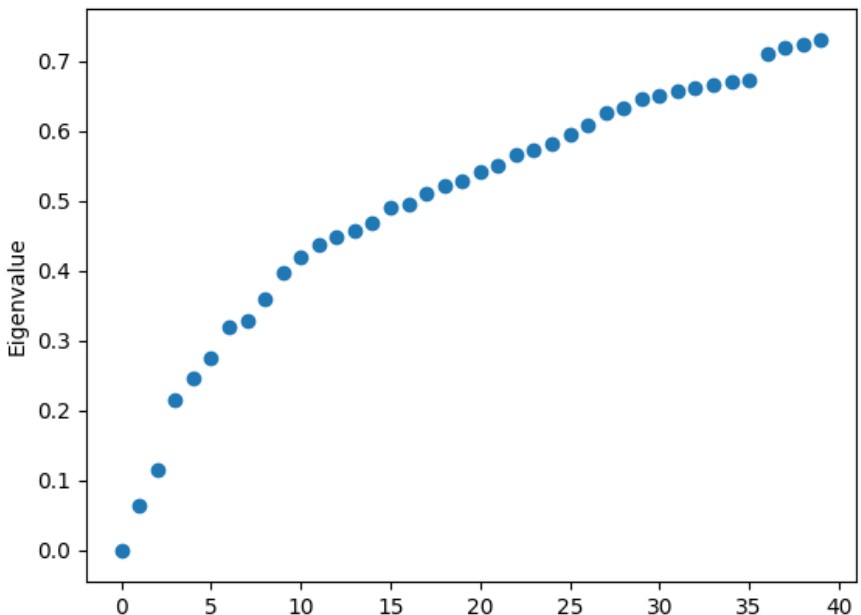

Figure 7: Scatter plot of eigenvalues to determine number of eigenvectors and clusters in spectral clustering

Table 6: Result for different embedding size (In this experiment, XGBoost was used as a classifier and bert-prob was used along with emdeddings of different sizes as independent variable. No other feature apart from these was used)

| SkillBERT embedding size | Precision | | Recall | | F1-score | |
| --- | --- | --- | --- | --- | --- | --- |
| | Class 0 | Class 1 | Class 0 | Class 1 | Class 0 | Class 1 |
| 32 | 98.12% | 91.65% | 95.47% | 80.12% | 96.78% | 85.50% |
| 64 | 98.32% | 91.80% | 97.26% | 81.10% | 97.79% | 86.12% |
| 128 | 99.12% | 92.65% | 97.47% | 83.80% | 98.29% | 88.00% |
| 256 | 99.12% | 92.56% | 97.40% | 83.79% | 98.25% | 87.96% |

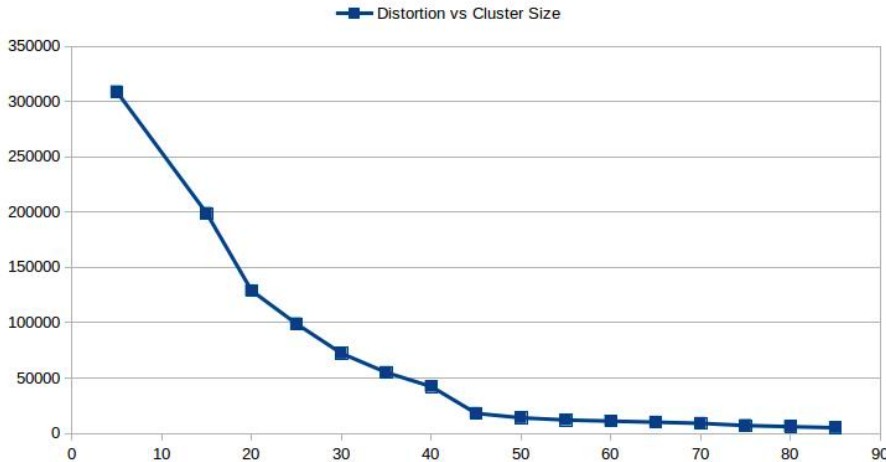

Figure 8: Elbow method graph to determine the number of clusters in K-means clustering

Table 7: Examples of some candidate and job profiles

| Candidate or Job | Skill set | Probable competency groups |
|---|---|---|
| Candidate1 | Design, knockoutjs, corel draw | Tool design, mechanical design, front end, web development |
| Candidate2 | Statistical modeling, statistical process control | Statistics, production operations |
| Job1 | Analytical skills, project execution, accounting | Financial operations, business analytics, statistics, accounts |
| Job2 | Digital marketing, cash management, ms office, ms excel, ms word, tally | Taxation, banking, statistics |

### A.3 SKILLBERT TRAINING

The dataset used for training the SkillBERT model can be downloaded from here. It contains the list of skills present in job requisitions. Table 7 contains examples of some candidate and job profiles. We leveraged Bert-Base architecture on the job-skill data to generate embeddings of size 768, details of it can be found here. Finally, the embeddings generated using the SkillBERT model can be downloaded here.

### A.4 ANNOTATED DATA PREPARATION

We had domain experts related to each field for manual annotation of competency group to a skill for generating the training dataset. Following instructions were given to them:

1. A single skill can belong to multiple groups

2. If they don't know about a particular skill then they can google it and based on that information can annotate that skill

3. If a skill is related to a particular group then they have to further classify it as a core(strongly related) or fringe(weakly related) skill to that particular group

The mapping of competency groups and skills can be downloaded here.

Table 8: Feature Description

| Feature Name | Feature Type | Dimensionality |
|---|---|---|
| bert_0 - bert_127 | SkillBERT Embedding | 128 |
| bert-prob | SkillBERT Embedding | 1 |
| 0-34 | Spectral clustering label | 35 |
| value1-value3 | skill-skill similarity | 3 |
| tf-idf | TFIDF | 1 |
| bert_grp_sim | skill-group similarity | 1 |
| core_skill_count,fringe_skill _count | group based feature | 2 |

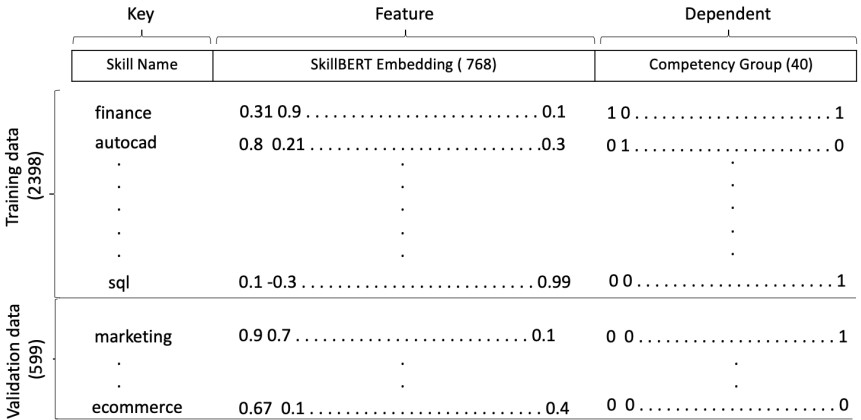

Figure 9: Data format used for creating bert_ prob feature

## A.5 FEATURES

The details of features used in the training of Bi-LSTM model, which gave us the best performance are given in Table 8 .

## A.6 TRAINING DATA FORMAT

The data format used for generating bert_ prob feature and final model is shown in Figure 9 and Figure 10 respectively.

## A.7 RUNNING THE EXPERIMENT

The code to run all the experiments mentioned in the paper can be downloaded here. This codebase uses python 3.6 and all the packages used for this experiment can be downloaded by installing requirements.txt. An overview of all the folders present in the code is given below:

**1. training_codes:** This folder contains the main python files used for running the experiments mentioned in the paper. Inside the main() method there are functions for data preparation, training, and testing. We have provided comments in each section for better understating of the modules. The code present in the file "skillbert_spectral_clustering.py" is used to train Bi-LSTM model on SkillBERT and spectral clustering related features which gave us the best performance. You can directly jump to this code if you don't want to run other intermediary experiments. The experiment for classifying a skill into core and fringe can be run using 3_class_classifier.py.

Apart from these if you want to run other experiments mentioned in the paper, you can do so by running "word2vec_only.py" for classifying skills using only Word2vec model, "skillbert.py" for classifying skills using only SkillBERT model, "bert_pretrain_only.py" for classifying skills using

| | Key | Feature | | | | | | | | Dependent |
|---|---|---|---|---|---|---|---|---|---|---|
| | Skill Name(2997) X competency Group(40) | SkillBERT Embedding ( 128) | bert_prob | spectral cluster index | TFIDF | bert_grp_sim | skill-skill similarity(3) | fringe_skill_ count | core_skill_count | 0/1 |
| ppc, digital marketing | 0.31 ..0.1 | 1 | 1 | 0.4 | 0.97 | 0.97,0.85,0.8 | 10 | 20 | 1 |
| ppc, finance | 0.31 ..0.1 | 0 | 1 | 0.4 | 0.63 | 0.63,0.5,0.4 | 3 | 15 | 0 |
| . | . | . | . | . | . | . | . | . | . |
| . | . | . | . | . | . | . | . | . | . |
| . | . | . | . | . | . | . | . | . | . |
| . | . | . | . | . | . | . | . | . | . |
| . | . | . | . | . | . | . | . | . | . |
| pig, big data | 0.11 ..0.2 | 1 | 5 | 0.2 | 0.89 | 0.89,0.82,0.6 | 5 | 12 | 1 |
| html, web development | 0.91 ..0.01 | 0.9 | 4 | 0.2 | 0.75 | 0.91,0.75,0.8 | 3 | 14 | 1 |
| . | . | . | . | . | . | . | . | . | . |
| . | . | . | . | . | . | . | . | . | . |
| nlp, machine learning | 0.22 ..0.1 | 0.8 | 9 | 0.7 | 0.78 | 0.8,0.79,0.7 | 9 | 25 | 1 |

*(Left row-group labels: "Training data(~96K)" spans the first rows through "pig, big data"; "Validation data (~24K)" spans "html, web development" through "nlp, machine learning".)*

Figure 10: Data format used for final model creation

only pre-trained BERT model and "skillbert_and_kmeans.py" for classifying skills using SkillBERT and k-means on SkillBERT embedding.

**2. feature_creation:** This folder contains the code for creating features used for training the models. If you don't want to go through each code, features created using these code files are already available in the feature_data folder. Codes present in the training_code also uses these CSV files directly for the model training.

**3. feature_data:** As mentioned before, this folder contains CSV files of features generated using codes present in feature_creation folder.

**4. model:** This folder contains the final model trained using all the experiments mentioned in the paper. Folder "skill_bert_spectral_clustering" contains the Bi-LSTM model which has been used as the final model.

**5. dataset:** This folder contains the final training and testing data used for each experiment. You can use these files to directly test the corresponding model.

