# OpenReview forum: "SkillBERT: “Skilling” the BERT to classify skills!"
_ICLR.cc/2021/Conference — Reject_

### Official Review · AnonReviewer3 · 2020-10-22
**Official Blind Review #3**

**Rating:** 6
**Confidence:** 4

**Review:**

This paper studied the problem of classifying skills into competency groups. The authors proposed SkillBERT, a BERT-based model, to extract the embeddings of skills and use that for the classification task.

Strengths:
1. The authors presented the details of feature engineering and experiment design. They also conducted extensive and comprehensive experiments which compare a lot of classification/feature engineering methods. It is a very good practical guide for doing related tasks.

2. The authors released the code and dataset, which largely improve the reproducibility of this work.

3. The paper is well written and easy to follow.

Weakness:
1. This paper studied a very specific problem which might only be interested to a very small group of researchers. It seems more like an industry track paper.

2. If I am correct, the problem size is relatively small. The total number of skills is 2997, and each of them will be classified into one or more competency group out of 40 competency group. Do we really a BERT model for this problem? Maybe the authors could provide some cost-performance tradeoff analysis here.

Questions for authors:
1. In Table 3, I only see the precision/recall/F-1 score for class 0 and class 1. Where is the number for class 2? Did I misunderstand this experiment?

Overall comments:

This is a good paper and I personally support it to be accepted. However, I do think that it seems more like an industry track paper. This is not for me to decide, maybe Meta-reviewer could share some opinions about whether this paper is suitable to ICLR.

---

> ### Author Response · Authors · 2020-11-24
> **Addressing review for SkillBERT**
>
> In Table 3, I only see the precision/recall/F-1 score for class 0 and class 1. Where is the number for class 2? Did I misunderstand this experiment?
>
> In Table 3, we have shown the results of competency group classification where the task is to tag competency groups to a skill. Hence the results are only for class 0 and class 1. The results of core vs fringe skill classification, where we are trying to tag whether a skill is a core, fringe, or unrelated to a competency group are shown in table 4. It has the performance metrics for class 0, class 1, and class 2.

---

### Official Review · AnonReviewer1 · 2020-10-28
**SkillBERT: "Skilling" the BERT to Classify Skills, novelty low, an application case study.**

**Rating:** 4
**Confidence:** 5

**Review:**

This paper proposes a model for job application screening. Since there is no job-related dataset available, the authors manually assigned labels to a large job application dataset. A skill set (e.g., Apache Hadoop, Apache Pig, HTML, Javascript) is firstly extracted from the job dataset.  Then a competency group is constructed (e.g., big data, front-end) as the labels. The problem is then formulated as a multi-label classification problem. That is, given a skill (which may belong to multiple competency groups), the model has to predict its competency groups. The authors proposed to use BERT as the main model. Moreover, the authors use additional features like similarity-based and cluster-based features. The experimental results are good. We think it can help recruiters find a suitable applicant.

However, this paper is straightforward. Using BERT as a main model for text classification is a well-known technique, and many papers already applied BERTs in other domains like biomedicine and law. So we think the technical contribution of this paper is limited. Furthermore, some parts of this paper are not clearly explained. For example, the authors mentioned that they also use some features like frequency-based and group-based features, but did not find detailed descriptions of these two features.

One positive side of this paper is that the authors release a publicly available job application dataset.
Establishing a dataset is time-consuming, and requires a lot of human efforts. The dataset consists of 700,000 job requisitions, which is large enough. It is good that the authors are willing to share this dataset.

To sum up, this paper proposes a skill classification model for job screening, which is useful, but we think that the methodology and its technical contribution are not strong enough. It might not be qualified as a regular paper for ICLR.

SkillBERT: "Skilling" the BERT to Classify Skills

---

> ### Author Response · Authors · 2020-11-24
> **Addressing review for SkillBERT**
>
> The details of the frequency-based and group-based features are given in the subsection "CORE AND FRINGE SKILLS" and "SKILL TFIDF FEATURE" under the "FEATURE ENGINEERING" section.

---

### Official Review · AnonReviewer4 · 2020-10-31
**Review of the submission called SkillBERT: “Skilling” the BERT to classify skills!**

**Rating:** 4
**Confidence:** 5

**Review:**

The manuscript focuses on a trending topic of applying a Bidirectional Encoder Representations from Transformers (BERT)-based prediction model to a new domain. More precisely, it addresses classifying Electronic Recruitment Records (ERRs) with respect to job skills. Its contributions include, but are not limited to, (i) releasing a related de-identified ERR dataset to the public domain, (ii) introducing a BERT-based embedding model, called SkillBERT, to group skills present in this ERR dataset into as competency clusters, and (iii) giving experimental evidence of the obtained modelling gains.

However, I am not convinced that these experiments are sufficient to support accepting the manuscript for the following five main reasons:

First, the compared models and methods are somewhat old and elementary (e.g., word2vec and TFxIDF), thereby failing to capture the current advances.

Second, in my opinion, the data annotation process calls for clarifications. For example, what was the expertise of the expert annotators, how were research ethics and informed consenting assured in this process involving human annotators as study participants, and do the authors obtain the rights to release the dataset and/or annotated dataset?

Third, the overall experimental setting does not seem to be adequately captured and justified, and I am unable to find a description of the performed statistical significance testing.

Fourth, the manuscript demonstrates only a reasonable understanding of related work in applications to skill/competency demand and existing studies in relevant computational methods/models/data (see, e.g., https://www.aclweb.org/anthology/search/?q=%22job%22 for recent relevant papers from the ACL Anthology that are largely missing from the reference list).

Fifth, the manuscript should be edited more carefully by clarifying both its contributions and limitations in relation to related work; describing and justifying its methodology and experiments; moving the in-text citations from the abstract to the body text of the later sections; and enhancing the image readability.

In conclusion, the study is valuable but needs further work.

---

> ### Author Response · Authors · 2020-11-24
> **Addressing review for SkillBERT**
>
> Q1. First, the compared models and methods are somewhat old and elementary (e.g., word2vec and TFxIDF), thereby failing to capture the current advances.
> We have compared the SkillBERT model with word2vec as the idea was to compare the performance of the transformer and non-transformer based models trained on the domain-specific dataset.
> Next, we compared SkillBERT with BERT-Base as we wanted to show how a BERT model trained on domain-specific can improve the quality of embeddings produced even trained on a smaller corpus such as a dataset of 700,000 records. We did not do any comparison with the TFxIDF approach rather it was used for feature creation.
>
> Q2. Second, in my opinion, the data annotation process calls for clarifications. For example, what was the expertise of the expert annotators, how were research ethics and informed consenting assured in this process involving human annotators as study participants, and
> do the authors obtain the rights to release the dataset and/or annotated dataset?
> We used the services of 5 annotators who are domain experts across various industries and have experience ranging between 8 to 12 years. They have taken several interviews and gone through a vast variety of resumes.
> During the annotation process, each annotator cross-examined a random sample of annotated data. Cases which were ambiguous were kept separate and were reviewed again in consultation between the actual annotator and the reviewer. The data for which the annotator and the reviewer could not agree were removed from the dataset
> Yes, authors have the right to publish the dataset. We have made sure that we are not revealing any personal information and have deidentified the published dataset.
> Q3. Third, the overall experimental setting does not seem to be adequately captured and justified, and I am unable to find a description of the performed statistical significance testing.
> In the experiment section, we have mentioned the pairs of models which were compared and had a statistically significant difference in performance. The same has been highlighted in table 3 (by prepending a star in front of the rows being compared) containing the results of the experiment.
>
> Q4. Fourth, the manuscript demonstrates only a reasonable understanding of related work in applications to skill/competency demand and existing studies in relevant computational methods/models/data (see, e.g., https://www.aclweb.org/anthology/search/?q=%22job%22 for recent relevant papers from the ACL Anthology that are largely missing from the reference list).
> As suggested by the reviewer, we have added some more related works which are relevant to our research. Below are details of newly added related work -
>
> Bian, S.; Zhao, W. X.; Song, Y.; Zhang, T.; and Wen, J.-R.2019.   Domain  Adaptation  for  Person-Job  Fit  with  Transferable  Deep  Global  Match  Network
> Alabdulkareem et al, Science Advances-2018, Unpacking the polarization of workplace skills;
> Xu et. al, AAAI-2018, Measuring the Popularity of Job Skills in Recruitment Market: A Multi-Criteria Approach;
> Qin et al., SIGIR'2018, Enhancing Person-Job Fit for Talent Recruitment: An Ability-aware Neural Network Approach;
>
>
> Q5. Fifth, the manuscript should be edited more carefully by clarifying both its contributions and limitations in relation to related work; describing and justifying its methodology and experiments; moving the in-text citations from the abstract to the body text of the later sections, and enhancing the image readability.
> We have moved the in-text citations from the abstract and have enhanced the image quality.
>
> Also, we have added the below changes in the introduction section to highlight the contributions and limitations of our work -
>
> In this paper, we have experimented with the application of BERT on the job-skill data which we have not seen done in the past literature. We have proposed a competency group classifier, which primarily leverages: SkillBERT, which uses BERT architecture and is trained on the job-skill data from scratch to generate embeddings for skills. The SkillBERT embeddings, when used as a feature in the competency group classifier, outperformed the results achieved using embeddings of Word2vec and BERT-base models respectively. However, due to the evolving nature of the recruitment industry, there can still be some skills that may not be part of SkillBERT, and hence, some manual intervention will be required for these new skills.

---

### Decision · Program_Chairs · 2021-01-07
**Final Decision**

**Decision:**

Reject

**Comment:**

The authors propose an approach for the task of categorizing competencies in terms of worker skillsets. This is a potentially useful (if somewhat niche) task, and one strength here is a resource to support further research on the topic. However, the contribution here is limited: The methods considered are not new, and while the problem has some practical importance it does not seem likely to be of particular interest to the broader ICLR community.